# Diversity? Great for Most Just Less So for Me: How Cognitive Abstraction Affects Diversity Attitudes and Choices

**DOI:** 10.3390/bs15050585

**Published:** 2025-04-27

**Authors:** Claudia Toma, Ashli B. Carter, Katherine W. Phillips

**Affiliations:** 1Solvay Brussels School of Economics and Management, Université Libre de Bruxelles, 1050 Bruxelles, Belgium; 2Columbia Business School, Columbia University, New York, NY 10027, USA; ashli.carter@gsb.columbia.edu (A.B.C.);

**Keywords:** workplace diversity, diversity attitudes, construal level, social dominance orientation

## Abstract

Individuals’ decisions to promote or limit diversity in the workplace are ambivalent and may be influenced by their cognitive focus. Drawing from construal level theory, we test across five studies whether individuals are more supportive of diversity when diversity is thought of more abstractly versus concretely. Furthermore, we examined the salience of diversity pros and cons as the underlying mechanisms, as well as the role of egalitarian beliefs as a boundary condition for this discrepancy. We tested these hypotheses in five studies, which varied in samples, cognitive focus manipulation, and measures. Dutch and American individuals indicated more positive attitudes toward diversity (Studies 1 and 2) and made more choices that enhanced diversity (Studies 2 and 5) when they considered diversity abstractly (for most companies and teams) rather than concretely (for their own company and teams). Furthermore, the discrepancy in diversity attitudes by construal level was more pronounced among individuals with egalitarian beliefs (Study 3) and was driven by the heightened salience of diversity pros at more abstract versus concrete levels of construal (Study 4). This research contributes to further understanding the ambivalent view of diversity and provides concrete recommendations for diversity management in organizations.

## 1. Introduction

Many organizations strongly endorse diversity within public-facing statements of their company’s values and commitments and have implemented numerous diversity practices ([10]). In fact, most Fortune 500 companies devote a page on their websites to highlighting their commitment to diversity ([25]; [44]). However, organizations often fall short of their stated goals to promote and include women and racial minorities within their ranks and, more generally, to have a positive impact on organizations ([26]; [10]). This discrepancy may occur, in part, due to the attitudes and choices of individual decision-makers to either promote or limit diversity within their work contexts. For many years, employees have been exposed to organizations’ public statements that exclusively focus on the positive aspects of diversity ([7]). This might contrast with individuals’ own views of diversity that are more likely to be ambivalent, seeing diversity as both a source of potential strength and weakness or even a threat ([1]; [41]; [22]). Within organizations, individuals’ attitudes and choices may not only be influenced by a desire to reap the benefits of diversity but also by concerns about managing diversity’s challenges ([17]). From an ideal and broad perspective, diversity appears to be a highly desirable goal, but its feasibility remains limited when individuals are asked to implement diversity concretely in their teams and organizations ([23]). Thus, advancing diversity is viewed as a complex goal with many benefits for organizations but also with many drawbacks and ideological pushbacks.

It is, therefore, key for management scholars and practitioners to understand individuals’ ambivalence toward diversity and how this could influence their attitudes and choices. We argue that the degree to which individuals consider the positive versus negative components of diversity and, as a result, form more or less favorable attitudes and choices in support of diversity will depend on individuals’ cognitive focus. We base this on a cognitive framework developed in social psychology ([31]; [39]; [46], [47]) that has been applied to understanding a wide range of organizational phenomena (for a review, see [52]). We argue that individuals have more favorable attitudes and choices about diversity when considering diversity more abstractly. In contrast, when considering diversity more concretely, we predict individuals’ attitudes and choices to be less favorable. This asymmetry occurs because individuals focus more on the pros of diversity at a more abstract level (when they think, for example, about teams and organizations in broad and distant terms), whereas individuals focus less on the pros and more on the cons associated with diversity at a more concrete level (when they think, for example, about their teams in specific and close terms). Importantly, we expect to see gaps between abstract and concrete considerations of diversity more among individuals who believe in the merits of diversity—those with egalitarian beliefs—than those with less positive diversity beliefs. We reason this will occur because individuals must be open to arguments in favor of diversity for these to matter more in the abstract than in the concrete. By taking people’s egalitarian beliefs into account, we acknowledge that people’s cognitive focus might interact with their ideological beliefs to explain support for diversity.

Our research builds and extends existing research about diversity ambivalence ([23]; [24]) and the role of egalitarianism in diversity judgments ([2]; [51]). By utilizing a construal level framework to understand individuals’ attitudes and choices about workplace diversity, we contribute to understanding why advancing diversity within organizations remains a pervasive challenge. More specifically, we replicate existing findings about the role of social distance in diversity choices ([23]) while providing evidence of the underlying mechanism of this ambivalence, namely the difference in focus on diversity pros and cons triggered by the cognitive focus. In addition, we show that people’s support for diversity is reinforced by egalitarian beliefs, but this only occurs when people’s cognitive focus is abstract, not concrete. Taken together, we identify a cognitive mechanism for why individuals who *generally* support the aims of diversity may fail to do so within their own organizational or team contexts. This has implications for people’s support toward diversity initiatives and suggests that people’s attitudinal and behavioral support is hard to obtain ([24]).

### 1.1. Ambivalent Views of Diversity

Although diversity is often touted uniformly as a virtue and strength at the organizational level of most modern companies, individuals’ attitudes are often more complex (for a review, see [38]). Indeed, over the last two decades, research within the management literature has demonstrated largely “mixed” effects of diversity, whereby diversity operates in the workplace with both positive and negative outcomes (see [6]; [17]; [53]; for reviews). Hence, diversity has often been referred to as a “double-edged sword”. On the one hand, having individuals who bring different perspectives and backgrounds reflects the commitment to fairness and equality, enhances decision-making, and helps organizations to more effectively meet the needs of diverse clientele ([6]; [13]). On the other hand, by introducing differences in backgrounds and perspectives, diversity can increase interpersonal conflict and detract from work group functioning by diminishing trust, communication, and coordination ([3]; [27]; [28]; [34]). For example, individuals associate diversity with less power ([45]) and less successful cooperation among team members ([43]).

Within organizations, individuals may hold ambivalent views of diversity to the extent that they have experienced both positive and negative aspects of diversity practices. Looking at the current state of diversity practices from an inter-group and organizational justice perspective, we see that advanced and disadvantaged employees have ambivalent views about their effectiveness. While diversity initiatives aim to tackle inequalities and the discrimination of disadvantaged employees ([30]), they rarely benefit from diversity initiatives ([26]). At the same time, employees from advantaged groups often feel threatened and resist them ([12], [11]; [22]), even if they acknowledge that diversity brings opportunities such as enriched learning from different cultures, economic growth, and innovation ([41]).

Therefore, we reason that diversity’s beneficial outcomes and various liabilities may complicate the degree to which individuals value and make decisions that promote workplace diversity. In essence, individuals may view advancing diversity as somewhat risky and, as a result, might assess whether the opportunities are worth the challenges. Indeed, previous work has shown that when allowed to select future group members, individuals prefer to work with racially similar rather than with dissimilar others in part because they value predictability ([20]). We argue that understanding how individuals weigh positive and negative factors associated with diversity is critical as individuals’ attitudes and choices surrounding diversity are vital components to ensuring that organizations can actually reap the benefits of their diversity (e.g., [13]; [21]). We draw upon insights from construal level theory to propose that individuals’ attitudes and choices to support diversity depend on their cognitive focus.

### 1.2. Construal Level and Weighing the Pros and Cons of Actions

According to construal level theory, the way in which individuals “construe” or come to understand behavior—either more abstractly or more concretely—influences the consideration of and relative weight placed on arguments for and against pursuing a course of action ([46], [47]). When construing more abstractly, individuals think about why and whether they will pursue a goal in the first place, focusing primarily on an action’s associated benefits and what, if anything, is to be gained. When construing more concretely, however, individuals give more weight to and are more influenced by more secondary concerns, such as whether a desirable action also has potential drawbacks (e.g., [14]), thus paying attention to the details and case-specific features of a given target within a particular context (e.g., [5]). Hence, abstract construals are simple, decontextualized representations that focus attention on the primary features of a given target, whereas concrete construals are more complex, contextualized representations that shift attention to the more secondary features of a target.

This hierarchical structure of abstract and concrete mental representations provides important insight into how individuals evaluate and weigh the pros and cons of potential actions. The consideration of the pros of a given action is superordinate to the consideration of the cons of that same action. This is because arguments against pursuing an action do not need to be considered when arguments in favor of an action are not sufficiently present ([14]; [16]). At a more abstract level of construal, pros are more salient, whereas at a more concrete level of construal, cons are more salient. Individuals form more positive attitudes when construing targets at a more abstract level ([54]) because they represent objects that are more psychologically distant (i.e., temporally, socially, or spatially) with more pros and those that are psychologically close with more cons. For example, individuals are more supportive of social policies that are to be implemented in the more distant (versus near) future. For instance, this effect is driven by the relative ease with which individuals generate policy pros (versus cons) when construing more abstractly (versus concretely; [19]; [40]). Similarly, people exposed to abstract statements about the value of diversity are more inclined to choose diverse teams than when they are not exposed ([8]). The study of [23] ([23]) is particularly relevant to the current research, which showed that individuals prefer diverse teams in a socially distant situation due to the stronger impact of pro-diversity desirability concerns. In contrast, individuals prefer similar team members in a close social situation due to a stronger weighing of anti-diversity feasibility concerns. We extend the work of [23] ([23]) by providing evidence of the focus on pros vs. cons under different construals and, more importantly, the role of egalitarian beliefs in shaping the impact of construals.

### 1.3. Construal Level and Motivated Cognitions of Diversity

Within the management literature in particular, attention has been placed on how individuals’ values and beliefs motivate broader versus more narrow definitions of diversity. For example, individuals with anti-egalitarian beliefs and those who identify strongly with their White racial group are more likely to define diversity broadly as inclusive of non-demographic dimensions (i.e., occupation) to maintain racial inequality and justify opposition to affirmative action policies ([49]; [48]). At the same time, white participants with anti-egalitarian beliefs endorse more narrow definitions of diversity in a way that includes fewer disadvantaged demographic groups ([57]).

We build on this work by investigating broad and narrow definitions of diversity to examine how more *abstract* versus *concrete* construals of diversity shape attitudes and choices in support of diversity and also as a function of their egalitarian beliefs. Anti-egalitarian beliefs were found to operate in conjunction with other cognitive processes to guide egalitarian efforts, such as support for diversity ([51]). Indeed, shifting weights between diversity’s primary function (i.e., pros) and secondary concerns (i.e., cons) at abstract and concrete levels of construal is predicated on the underlying belief that diversity is beneficial in the first place. In other words, egalitarian beliefs might have a moderating role as individuals must be open to the positives of diversity for these pros to matter more in the abstract than in the concrete. For this reason, we propose that asymmetries in individuals’ positive diversity attitudes and choices between more abstract versus concrete construals of diversity will occur most strongly for individuals with egalitarian beliefs. For these individuals, an abstract consideration of diversity should highlight the benefits of diversity relative to its drawbacks, whereas a more concrete consideration will bring greater attention to potential downsides in a way that negatively impacts attitudes and choices in support of diversity. This also suggests that the role of anti-egalitarian beliefs should be stronger in the abstract than in the concrete construal, given that the abstract focus leads to more inclusive categorization, support for multiculturalism, and more pro-social behaviors, which correspond to what people with egalitarian beliefs generally support ([33]; [56]). In contrast, individuals with anti-egalitarian beliefs should consider diversity’s primary function (i.e., arguments in favor of diversity) as insufficient to garner further consideration of secondary concerns (i.e., arguments against diversity) regardless of whether they construe diversity more abstractly or concretely. As a result, anti-egalitarian individuals’ attitudes and choices towards diversity should remain relatively stable across different construal levels. Taken together, we integrate insights from construal level theory with those regarding motivated cognitions of diversity to better understand the potential challenges organizations may face in *actually* advancing diversity despite having employees who may *generally* support diversity’s aims.

### 1.4. Hypotheses and Current Studies

Our central hypothesis is that individuals will form more favorable attitudes and make more choices in support of diversity when they construe diversity more abstractly compared to when they construe diversity more concretely due to the greater salience of diversity pros versus cons at more abstract levels of construal, an effect further moderated by individuals’ egalitarian beliefs. We tested this hypothesis in five studies in which we operationalize abstract versus concrete construals of diversity as individuals’ consideration of diversity’s impact at an aggregated level for most organizations and teams versus their consideration of diversity’s impact at a case-specific level within their own organization and teams. Our studies build on each other, starting with simpler designs testing the central hypothesis (the effect of construal on diversity attitudes—Studies 1 and 2—and choices—Study 2) to more complex designs testing its boundary condition (the moderating role of egalitarian beliefs—Studies 3, 4, and 5) and its underlying mechanism (the mediating role of pros and cons—Studies 4 and 5).

More specifically, in Study 1, we assess individuals’ attitudes about diversity’s value either for most groups or for their groups in a Dutch student sample. Study 2 examines choices to promote, as well as positive attitudes toward diversity, with Dutch employees. In Study 3, we investigate whether (anti)egalitarian beliefs moderate the impact of abstract versus concrete construal level on positive diversity attitudes within a White American working sample. In Study 4, again using a White American working sample, we test the salience of diversity pros versus cons as a possible mechanism for differences in favorable attitudes towards diversity at differing levels of construal, as well as assess the moderating role of individuals’ (anti)egalitarian beliefs. Finally, in Study 5, using a White American working sample, we examine whether the relative salience of diversity pros versus cons at differing levels of construal also drives choices to promote diversity and whether (anti)egalitarian beliefs moderate the effect of construal level on diversity choices. Because each study tested the main hypothesis gradually using a different variable, we formally hypothesize the following (see Table 1 for an overview of studies, protocols, and tested hypotheses).

**Hypothesis 1.** 
*Individuals will indicate more positive diversity attitudes when diversity is construed more abstractly compared to when diversity is construed more concretely (a).*


**Hypothesis 2.** 
*Individuals will make more diversity choices when diversity is construed more abstractly compared to when diversity is construed more concretely (a) and especially when they hold egalitarian beliefs (b).*


**Hypothesis 3.** 
*The relative salience of diversity pros versus cons will be greater when diversity is construed more abstractly compared to when diversity is construed more concretely (a) and especially for individuals with egalitarian beliefs (b).*


**Hypothesis 4.** 
*Greater salience of diversity pros versus diversity cons will mediate*
*the relationships between construal level and diversity attitudes and choices, (a) and is further moderated by (anti)egalitarian beliefs (b).*


## 2. Study 1

Our sample included sixty-one Dutch students (16 male; *M*_age_ = 20.82, *SD*_age_ = 2.28) who were recruited to participate for credit hours or monetary compensation from Tilburg University. The target sample was calculated using G*power 3 based on the effect size of [23] ([23]). For a *d* = 0.66, alpha level set at 0.05, and a power of 0.80, the suggested sample size was 60 participants. Participants were randomly assigned to either an abstract or concrete construal condition in a between-subjects design.

### 2.1. Procedure

Participants were recruited on campus and asked if they would participate in a short study. If participants accepted, the researcher gave them a printed copy of the survey and left them alone to complete it. Participants were told that we were interested in people’s perceptions about groups. Participants in the abstract construal condition answered questions assessing their attitudes about diversity’s value for most groups, while participants in the concrete construal condition answered the same questions, but the focus was on their own groups. For additional measures collected for exploratory purposes and full-scale items, please see the Appendix A.

#### Positive Diversity Attitudes Measure

Participants indicated their attitudes about diversity’s value in groups with the exact wording of the question depending on the condition. Sample items include, “I believe that [*most groups*/*groups to which I belong*] benefit from the involvement of people with different backgrounds”; “I think that for [*most groups*/*groups to which I belong*] having members with different backgrounds is helpful”; and “I think that [*most groups*/*groups to which I belong*] should be characterized by members’ diversity” (six items) on a seven-point scale, from 1 = *not at all* to 7 = *very much*. Scale reliabilities, as well as means, standard deviations, and correlations among studied variables across all studies, are reported in Table 2.

### 2.2. Results

To examine whether positive attitudes towards diversity differed by construal level, we conducted an independent-sample *t*-test assessing mean differences between the abstract and concrete construal conditions. As predicted, individuals indicated more positive attitudes about diversity for most groups (*M* = 5.16, *SD* = 0.82) than for groups to which they belonged (*M* = 4.43, *SD* = 0.83), *t*(59) = 3.44, *p* = 0.001, *d* = 0.89, in support of Hypothesis 1a.

### 2.3. Discussion

In Study 1, we found initial support for our predictions. More specifically, we found that individuals indicated more positive attitudes towards diversity when they considered diversity’s value more abstractly for most groups compared to when they construed diversity more concretely for their own groups. However, one of the limitations of this study is that ‘groups’ is a large concept, and we cannot be totally sure that participants thought of comparable groups in both conditions. To provide a more conservative test of our hypothesis, we tested in our second study whether the same pattern emerged within a sample of working adults asked about their working teams and who may be more familiar with both the positives and negatives of working in diverse teams. Furthermore, we tested whether the difference in support for diversity between abstract and concrete construals would extend to individuals’ choices.

## 3. Study 2

Our sample included seventy Dutch employees (33 female; *M*_age_ = 35.20, *SD*_age_ = 4.47) from various national and international companies in the Netherlands. Contact was established with an employee of a certain company who then agreed to distribute the survey amongst their colleagues. All participants provided informed consent prior to participation. Participants were randomly assigned to either the abstract or concrete construal condition in a between-subjects design. Again, the target sample was calculated to be 60 participants.

### 3.1. Procedure

The survey was distributed to employees using the Qualtrics survey platform. Participants first indicated their attitudes about diversity. Participants in the abstract construal condition answered diversity attitude measures in reference to most companies and teams. Participants in the concrete construal condition answered these questions for their own companies and teams. Next, participants answered eight questions referring to their gender (male—52.9%, female—47.1%), age category (20–35—58.6%, 36–45—21.4%, 46–55—10%, 56–65—10%, >65), academic degree (High school—10%, Bachelor’s—38%, Master’s—49%, PhD—4%, None), ethnicity (African American—4.5%, Arabic, Asian—4.5%, Caucasian—91%, Hispanic/Latino), culture group (American—4.5%, Arabic, Asian—2.9%, East European—2.9%, North-West European—83%, South European—7.1%, South American), political group (left-wing—52%, right-wing—48%), and two personality traits (extraverted—59% vs. introverted—41%; and agreeable 82% vs. disagreeable—18%). The answers to these questions were used to establish a profile for each participant.

Participants were then invited to choose four individuals from a list of eight potential group members to form a team. A profile for each potential group member mirrored the information provided by the participants and consisted of the same eight characteristics (i.e., gender, age, ethnicity, cultural group, level of education, political preference, extraversion, and agreeableness). For example, one potential member was described in the following manner:

Male, 65 years old, and with a high school degree. He is Caucasian and grew up in Scandinavia. He is a right-wing voter and would describe himself as a soft-spoken person who prefers to keep the peace over standing by his own opinion.

Participants in both conditions then selected four individuals for the team. The construal manipulation for diversity choices was based on [23] ([23]), who showed that abstract construal activates desirability concerns, while concrete construal activates feasibility concerns. Participants in the abstract condition were told to choose a mix of four people that most closely represent the way in which a company should do it (desirability). We decided, in this case, to reference “a company” rather than “most companies” as in our previous operationalizations to make the choice seem more realistic. Participants in the concrete condition were told to imagine that they had the final say in composing a team of people who would be their direct colleagues and to choose the four people with whom they would feel most comfortable working (feasibility). For the full study materials, see the Appendix A.

### 3.2. Measures

#### 3.2.1. Positive Diversity Attitudes

Participants indicated their attitudes about diversity’s value in companies and teams, again with the exact wording depending on the condition. In this study, we adapted items from the [50] ([50]) diversity beliefs questionnaire. Sample items include, “[*Most companies*/*The company I currently work for*] should hire people from different backgrounds”; “At [*most companies*/*the company I currently work for*], a mix of people from different backgrounds helps to do a task well.”; and “[*In most companies, teams*/*My teams at the company I currently work*] for can benefit from the involvement from people with different ethnic backgrounds.” (six items) on a seven-point scale, from 1 = *strongly disagree* to 7 = *strongly agree*.

#### 3.2.2. Diversity Choices

We created an index of participants’ choices to promote diversity by comparing participants’ characteristics with those of the group members they chose. Every dissimilar characteristic was scored +1, while every similar characteristic was scored −1, and then the scores were summed up (across all characteristics and for all chosen members). For example, if participants chose all four team members whose characteristics were all similar to their own, they would receive an overall diversity choice score of −32, whereas choosing four team members who differed from the participant on every dimension would yield an overall diversity choice score of +32.

### 3.3. Results

#### 3.3.1. Positive Diversity Attitudes

We first assessed whether positive diversity attitudes differed by construal level by conducting an independent-sample *t*-test assessing mean differences between the abstract and concrete construal conditions. The analysis confirmed that, as in our previous study and in support of Hypothesis 1a, individuals reported more positive diversity attitudes in the abstract (*M* = 5.71, *SD* = 0.64) than in the concrete (*M* = 5.29, *SD* = 0.92) construal condition, *t*(68) = 2.19, *p* = 0.032, *d* = 0.52, this time using a different measure of positive diversity attitudes.

#### 3.3.2. Diversity Choices

Next, we tested differences in individuals’ diversity choices in the abstract versus concrete construal conditions. Recall that overall diversity choice scores could range in theory from −32, being not at all diverse, to +32, being very diverse. Likewise, demographic and cognitive diversity choice scores could range from −16 to +16. Within our sample, we found that overall diversity choice scores ranged from −20 to −4, with demographic diversity choice scores ranging from −18 to −4, and cognitive diversity choice scores ranging from −8 to +3.

We predicted that individuals would make more choices that promoted diversity at a more abstract level of construal compared to a more concrete level of construal. An independent-sample *t*-test confirmed this prediction for overall diversity choices in the abstract (*M* = −12.19, *SD* = 3.75) versus concrete (*M* = −13.97, *SD* = 2.92) construal conditions, *t*(68) = 2.24, *p* = 0.029, *d* = 0.54, in support of Hypothesis 2a. Table 2 also offers correlations with demographic and cognitive diversity choices.

### 3.4. Discussion

In Study 2, we found additional evidence that individuals’ attitudes toward diversity are more positive when they think about diversity more abstractly rather than concretely, replicating our previous results within a Dutch working sample. In addition, we also demonstrated that individuals’ more abstract versus concrete construal of diversity also shapes individuals’ downstream choices to promote diversity. Indeed, individuals created more diverse teams at more abstract than concrete levels of construal.

In our next study, we wanted to build on these findings in a few ways. First, we wanted to examine whether the effect would generalize to White Americans. Moreover, we wanted to test the predicted moderating effect of egalitarian beliefs. Our theoretical reasoning suggests that individuals will have more positive attitudes when construing diversity abstractly because they will focus more on the primary function (i.e., the potential benefits) of diversity, whereas a more concrete construal of diversity will guide attention more to secondary concerns, such as diversity’s potential drawbacks. This suggests a boundary condition of our predicted effects, namely that individuals must a priori acknowledge or buy into the potential benefits of diversity (i.e., have egalitarian beliefs). Conversely, for individuals with anti-egalitarian beliefs, diversity should not be sufficiently desirable regardless of whether they construe diversity abstractly or concretely. As a result, these individuals’ attitudes towards diversity should not vary as a function of construal level. We tested this boundary condition in our third study by measuring individuals’ social dominance orientation ([35]), a common measure of anti-egalitarian beliefs.

## 4. Study 3

Our sample included ninety-five currently employed White Americans (55 male; *M*_age_ = 35.56, *SD*_age_ = 9.05) recruited through Amazon’s Mechanical Turk website ([4]). Participants were randomly assigned to either the abstract or concrete construal condition in a between-subjects design. Our design also included an additional experimental factor whereby we wanted to examine whether the effects of thinking of diversity more abstractly versus more concretely would be moderated by an independent manipulation of construal on an unrelated topic. This manipulation had no effect on our manipulation, and it is, therefore, not reported.

### 4.1. Procedure

Participants were told we were interested in how individuals think about groups and teams. Participants in the abstract construal condition answered questions assessing their attitudes about diversity’s value for most groups, and participants in the concrete construal condition answered questions assessing their attitudes about diversity’s value for their own groups. Afterwards, individuals indicated their (anti)egalitarian beliefs. For additional measures collected for exploratory purposes, as well as the full scale items and the attention check, please see the Appendix A.

### 4.2. Measures

#### 4.2.1. Positive Diversity Attitudes

Participants indicated their attitudes about diversity’s value with the same six items from Study 1 for either most groups or for groups to which they belong based on experimental conditions.

#### 4.2.2. Anti-Egalitarian Beliefs

Participants indicated their anti-egalitarian beliefs using the Social Dominance Orientation scale ([35]). Sample items include, “Some groups of people are just more worthy than others”; “It’s OK if some groups have more of a chance in life than others”; and “If certain groups of people stayed in their place, we would have fewer problems” (eight items) on a seven-point scale, from 1 = *strongly disagree* to 7 = *strongly agree*.

### 4.3. Results

Again, we assessed whether positive attitudes towards diversity differed by construal level by conducting an independent-sample *t*-test that assessed mean differences between the abstract and concrete construal conditions. As predicted, individuals indicated more positive attitudes about diversity for most groups (*M* = 5.63, *SD* = 1.23) than for groups to which they belonged (*M* = 5.11, *SD* = 0.93), *t*(93) = 2.28, *p* = 0.025, *d* = 0.48, providing additional support for Hypothesis 1a.

Next, we examined whether the influence of a more abstract versus concrete construal of diversity on positive diversity attitudes was moderated by anti-egalitarian beliefs. We expected the influence of construal to be stronger among individuals with lower (versus higher) anti-egalitarian beliefs. We reasoned this would occur because individuals with greater anti-egalitarian beliefs should view diversity as undesirable regardless of their construal level. To test this, we conducted a linear regression with main effects entered for construal level (abstract = 1, concrete = 0) and anti-egalitarian beliefs, as well as their two-way interaction term. As expected, anti-egalitarian beliefs moderated the influence of more abstract versus concrete construal on positive attitudes towards diversity in the predicted pattern (*B* = −0.32, *t*(94) = 2.57, *p* = 0.012). More specifically, the positive influence of abstract (versus concrete) construal on favorable diversity attitudes was stronger for those lower in anti-egalitarian beliefs, in support of Hypothesis 1b (see Figure 1).

### 4.4. Discussion

In Study 3, we replicated findings from our previous studies showing that individuals have more favorable attitudes toward diversity when they construe diversity more abstractly for most groups rather than concretely for their own groups, this time among White American employees. We also confirmed an important condition under which this effect is more likely to emerge. Individuals with lower anti-egalitarian beliefs (i.e., those with higher egalitarian beliefs) were more likely to show this discrepancy in their attitudes toward diversity. This provides greater support for our theoretical reasoning that at more abstract (versus concrete) levels of construal, individuals evaluate diversity in terms of its primary function (i.e., its benefits) rather than secondary concerns (i.e., potential drawbacks) and so should only be valued more positively at more abstract levels of construal for those who believe that diversity does, in fact, have benefits. However, we wanted to test this more directly. More specifically, in our next study, we tested whether diversity pros (versus cons) are more salient at more abstract (versus concrete) levels of construal among those with more egalitarian beliefs and whether this relative salience drives positive attitudes towards diversity.

## 5. Study 4

Our sample included 194 currently employed White Americans (111 male; *M*_age_ = 38.23, *SD*_age_ = 9.95) who were recruited through Amazon’s Mechanical Turk website. Participants were randomly assigned to either the abstract or concrete construal condition in a between-subjects design.

### 5.1. Procedure

Participants were again told that we were interested in how individuals think about groups and teams before answering questions assessing their attitudes about diversity’s value. Afterward, participants listed diversity pros and diversity cons (in counterbalanced order). Participants in the abstract construal condition answered diversity attitude measures and listed diversity pros and cons in reference to most groups and teams. Participants in the concrete construal condition answered these questions for their own groups and teams. Pros might include benefits to work productivity, such as creativity, or moral considerations of promoting equality and fairness between groups. Cons could include feelings of exclusion from majority group members, concerns that greater diversity will adversely change workplace culture, or apprehensions that increasing workplace diversity will diminish organizational standards and competency levels.

Finally, all participants indicated their anti-egalitarian beliefs. For additional measures collected for exploratory purposes, as well as the full scale items and the attention check, please see the Appendix A.

### 5.2. Measures

#### 5.2.1. Positive Diversity Attitudes

Participants indicated their attitudes about diversity’s value with the same six items as in Study 1, with a slight modification. All items were changed to reference “groups and teams” rather than “groups” but were otherwise identical. Again, participants answered questions about either most groups and teams or for groups and teams to which they belong based on experimental conditions.

#### 5.2.2. Diversity Pros and Cons

In a free-response format with ten blank text-entry fields, participants were instructed to list as many pros and as many cons (in counterbalanced order) that came to mind when they thought of diversity, either in most groups and teams or in groups and teams to which they belong, depending on the experimental condition. Participants were required to list at least one pro and one con.

#### 5.2.3. Anti-Egalitarian Beliefs

Participants indicated their anti-egalitarian beliefs using the same scale as in Study 3. The order in which these variables were measured was counterbalanced.

### 5.3. Results

#### 5.3.1. Positive Diversity Attitudes

Again, we assessed whether positive attitudes towards diversity differed by construal level by conducting an independent-sample *t*-test that assessed mean differences between the abstract and concrete construal conditions. Contrary to predictions and our previous findings, individuals’ positive attitudes towards diversity did not differ between the abstract and concrete construal conditions, *p* = 0.24. Additionally, a linear regression predicting positive diversity attitudes with main effects entered for construal level (abstract = 1, concrete = 0) and anti-egalitarian beliefs, as well as their two-way interaction term, did not reveal the expected interaction, *p* = 0.86.

#### 5.3.2. Diversity Pros and Cons

We next assessed whether the relative salience of diversity pros versus cons varied by construal level. To do so, we first had an independent coder, who was blind to the conditions, read participants’ lists of pros and cons to count the valid number of pros (i.e., positive diversity-related factors) and cons (i.e., negative diversity-related factors). See Table 1 for means, standard deviations, and correlations with other variables.

We expected the salience of diversity pros (versus cons) to be greater in the abstract (versus concrete) construal condition. An independent-sample *t*-test of mean differences in diversity pros between the abstract (*M* = 4.36, *SD* = 2.19) and concrete (*M* = 3.87, *SD* = 1.71) construal conditions confirmed the expectation that individuals generated more pros at a more abstract level of construal, at a level of marginal significance, in support of Hypothesis 3a, *t*(192) = 1.74, *p* = 0.08, *d* = 0.25. However, an independent-sample *t*-test did not show any differences in the number of cons generated by the construal condition, *p* = 0.95.

We also tested whether the salience of diversity pros versus cons in the abstract versus concrete conditions was further moderated by anti-egalitarian beliefs. A linear regression predicting diversity pros with main effects entered for construal level (abstract = 1, concrete = 0) and anti-egalitarian beliefs, as well as the two-way interaction term between construal level and anti-egalitarian beliefs, revealed a significant interaction in the expected direction (*B* = −0.29, *t*(193) = 2.15, *p* = 0.035). More specifically, the greater salience of diversity pros in the abstract (versus concrete) construal condition was stronger for those with lower anti-egalitarian beliefs (see Figure 2), supporting Hypothesis 3b. Using the same linear regression model to predict the number of diversity cons did not reveal a significant interaction between the construal condition and anti-egalitarian beliefs, *p* = 0.76.

#### 5.3.3. Mediation Analysis

Although we did not find a direct effect of the construal level, or the interaction of construal level and anti-egalitarian beliefs, on positive diversity attitudes in this particular study, we still wanted to assess whether there may be an indirect effect of construal level on positive diversity attitudes through the relative salience of diversity pros at lower levels of anti-egalitarian attitudes. To do so, we used the SPSS29 macro designed by [18] ([18]) for mediated moderation bootstrapping analyses (Model 7) with the number of diversity cons generated entered as a covariate, creating 5000 bootstrap samples by randomly sampling observations with replacements from the original dataset (see bootstrapping method outlined by [36]). Ninety-five percent confidence intervals showed a significant indirect effect of construal level on positive diversity attitudes through the relative salience of diversity pros at the 16th (95% CI = [0.13, 0.51]) and 50th (95% CI = [0.06, 0.36]) percentiles of anti-egalitarian beliefs. The indirect effect was no longer significant at higher levels of anti-egalitarian beliefs (84th percentile, 95% CI = [−0.23, 0.20]). These analyses show a significant indirect effect of construal level on positive diversity attitudes occurring through the relative salience of diversity pros, but only at lower levels of anti-egalitarian beliefs in support of Hypothesis 4b, as shown in Figure 3. Because we did not find any influence of our construal manipulation or its interaction with anti-egalitarian beliefs for the number of diversity cons generated, we did not conduct further mediational analyses with this variable.

### 5.4. Discussion

Study 4 provided support for our proposed mechanism. Individuals who construed diversity more abstractly (versus concretely) paid greater attention to the pros associated with diversity, particularly when they were low in anti-egalitarian beliefs. Furthermore, although there was no direct effect of construal level on positive diversity attitudes in this particular study, we did find support for a significant indirect effect between construal level and positive diversity attitudes occurring through the relative salience of diversity pros for those low but not high in anti-egalitarian beliefs. Interestingly, the relative salience of diversity cons did not differ by construal level. Taken together, findings from Study 4 support our theoretical reasoning that attitudes toward diversity are more positive in the abstract rather than the concrete due to the heightened salience of potential benefits of diversity. In our final study, we tested whether individuals’ choices to promote diversity are also driven by the relative salience of diversity pros versus cons, as well as examining the moderating impact of anti-egalitarian beliefs on diversity choices.

## 6. Study 5

Our sample included 168 currently employed White Americans (85 male) who were recruited through Amazon’s Mechanical Turk website. Participants were randomly assigned to either the abstract or concrete construal condition in a between-subjects design.

### 6.1. Procedure

Participants were told that we were interested in studying groups and teams in the workplace before answering questions assessing their attitudes about diversity’s value. Next, participants answered eight questions (i.e., age, gender, ethnicity, culture, education, political preference, extraversion, and agreeableness) and then chose four individuals from a list of eight profiles in order to form a team, as in Study 2. Afterward, participants listed diversity pros and diversity cons (in counterbalanced order). Participants in the abstract construal condition answered diversity attitude measures, as well as listed diversity pros and cons in reference to most groups and teams, and were told to choose group members that most closely represent the way in which a company should do it. Participants in the concrete construal condition answered questions about diversity’s value, pros, and cons in reference to their own groups and teams and were asked to choose group members that they would feel most comfortable working with. Finally, all participants indicated their anti-egalitarian beliefs.

### 6.2. Measures

#### 6.2.1. Positive Diversity Attitudes

Participants indicated their attitudes about diversity’s value in companies and teams, again with the exact wording depending on the condition and using the same scale as in Study 2.

#### 6.2.2. Diversity Choices

We created an index of participants’ choices to promote diversity by comparing participants’ characteristics with those of the group members they chose. Every dissimilar characteristic was scored +1, while every similar characteristic was scored −1, and then the scores were summed up (across all characteristics and for all chosen members). As in Study 2, we created three choice scores: an overall diversity choice score, a demographic diversity choice score (i.e., age, gender, ethnicity, and culture), and a cognitive diversity choice score (i.e., culture, education, political preference, extraversion, and agreeableness).

#### 6.2.3. Diversity Pros and Cons

As in Study 4, in a free-response format, participants were instructed to list as many pros and cons (in counterbalanced order) that came to mind when they thought of diversity either in most groups and teams or in groups and teams to which they belong, depending on experimental conditions.

#### 6.2.4. Anti-Egalitarian Beliefs

Participants indicated their anti-egalitarian beliefs using the same scale as in previous studies. The order in which those variables were measured was counterbalanced.

### 6.3. Results

#### 6.3.1. Positive Diversity Attitudes

We first assessed whether positive diversity attitudes differed by construal level by conducting an independent-sample *t*-test assessing mean differences between the abstract and concrete construal conditions. Contrary to predictions and our previous findings, individuals’ positive attitudes towards diversity did not differ between the abstract and concrete construal conditions, *p* = 0.60. Additionally, a linear regression predicting positive diversity attitudes with main effects entered for construal level (abstract = 1, concrete = 0) and anti-egalitarian beliefs, as well as their two-way interaction term, did not reveal the expected interaction, *p* = 0.44.

#### 6.3.2. Diversity Choices

Next, we tested individuals’ abstract versus concrete choices for promoting diversity. Recall that the higher the score, the more choices to promote diversity participants made. Within our sample, we found that overall diversity choice scores ranged from −4 to +18, with demographic diversity choice scores ranging from −2 to +14 and cognitive diversity choice scores ranging from −6 to +8. Because we sampled White Americans exclusively in Study 5, we expected a smaller range in demographic diversity choice scores, reflecting more pro-diversity choices, compared to our Dutch sample in Study 2. This is because any choice that included a non-White, non-American group member would receive +2 points along these dimensions. See the Appendix A for the full profiles of potential group members.

We predicted that individuals would make more choices that promote diversity at a more abstract level of construal compared to a more concrete level of construal. An independent-sample *t*-test confirmed this prediction for overall diversity choices in the abstract (*M* = 7.83, *SD* = 4.37) versus concrete (*M* = 5.95, *SD* = 4.37) construal conditions, *t*(166) = 2.78, *p* = 0.006, *d* = 0.43, in support of Hypothesis 2a.

Next, we conducted a series of linear regressions predicting choices to promote diversity (overall, demographic, and cognitive) with main effects entered for construal level (abstract = 1, concrete = 0) and anti-egalitarian beliefs, as well as their two-way interaction term. Contrary to Hypothesis 2b, we did not find the expected interaction for any of the diversity choice scores, all *p*s > 0.61.

#### 6.3.3. Diversity Pros and Cons

We next assessed whether the salience of diversity pros (versus cons) varied by construal level. As in Study 4, we first had an independent coder, who was blind to the conditions, read participants’ lists of pros and cons to count the valid number of pros (i.e., positive diversity-related factors) and cons (i.e., negative diversity-related factors). See Table 1 for means, standard deviations, and correlations with other variables.

We expected the salience of diversity pros (versus cons) to be greater in the abstract (versus concrete) construal condition. Independent-sample *t*-tests of mean differences between the abstract and concrete conditions did not show any mean differences for either diversity pros or diversity cons, *p*s > 0.67. Likewise, linear regressions predicting diversity pros and diversity cons, respectively, with main effects entered for construal level (abstract = 1, concrete = 0) and anti-egalitarian beliefs, diversity pros/cons entered as a control variable, and the two-way interaction term between construal level and anti-egalitarian beliefs, did not reveal the expected interaction pattern for either diversity pros or diversity cons, *p*s > 0.66.

### 6.4. Discussion

In Study 5, we found additional evidence that individuals make more choices to promote diversity, particularly along demographics relative to cognitive dimensions of difference, when thinking of diversity more abstractly rather than more concretely. However, this tendency occurred for both individuals with egalitarian and anti-egalitarian beliefs. We did not replicate our previous findings showing that individuals’ attitudes toward diversity are more positive or that the relative salience of diversity pros is greater when diversity is construed more abstractly compared to more concretely, nor were these outcomes influenced by the interaction between construal level and anti-egalitarian beliefs in this study.

## 7. General Discussion

### 7.1. Key Findings

In a series of five studies, we examined whether individuals have more positive attitudes towards and make more choices to promote diversity at a more abstract versus concrete construal level. We also tested a proposed mechanism for this asymmetry in diversity attitudes and choices—the greater salience of diversity pros versus cons at more abstract versus concrete levels of construal. Finally, we investigated for whom this discrepancy between abstract and concrete construals of diversity was most likely to emerge. We hypothesized that because the positive impact of the heightened salience of diversity pros versus cons at more abstract levels of construal relies on the belief that diversity can be beneficial, individuals with more egalitarian beliefs would most likely show these effects.

We found support for many of our predictions. Across both Dutch and American samples, individuals did report more positive attitudes towards diversity when they thought about diversity at an abstract level for most companies and teams compared to when they thought about diversity more concretely within their own company and teams. Furthermore, White Americans with more egalitarian beliefs were more likely to show this asymmetry in positive diversity attitudes compared to their anti-egalitarian counterparts. More specifically, the positive influence of abstract (versus concrete) construal on favorable diversity attitudes was stronger for those lower in anti-egalitarian beliefs. In addition, individuals also made more diversity choices when they thought about diversity at an abstract level for most companies and teams compared to when they thought about diversity more concretely within their own company and teams. Importantly, the discrepancy between favorable attitudes towards diversity between abstract and concrete construals was driven by the greater salience of diversity’s pros at more abstract levels of construal.

### 7.2. Discrepancies Within and Between Studies

However, not all our predictions were supported consistently. For example, while the influence of construal level on individuals’ positive diversity attitudes was found in Studies 1 to 3, this was not the case for Studies 4 and 5. One of the explanations resides in the other variables concomitantly measured, notably the salience of diversity pros and cons. In addition, in Studies 4 and 5, the various dependent variables (diversity attitudes, diversity choices, and the pros and cons of diversity) and the moderator were measured in a counterbalanced way. Therefore, this additional variable could have complexified the design of those studies, where the influence of one variable is mixed with another. In Study 4, where diversity attitudes and diversity pros were measured concomitantly, the effect of construal of diversity pros might have diluted the effect on diversity attitudes. In other words, when we assessed the mediator, the effect of construal on the diversity attitudes disappeared, suggesting that the indirect impact might have canceled out the presence of the direct effect. In Study 5, where diversity choices were additionally measured, the effect of construal of diversity choices might have diluted the effect on diversity attitudes and diversity pros. This suggests a potential gradual effect of construal on these different measures, with the effect on diversity choices being more robust than the effect of diversity attitudes and pros (see also [23]).

In addition, while (anti)egalitarian beliefs moderated the influence of construal level on individuals’ positive diversity attitudes, this was not the case for choices promoting diversity, which were greater in the abstract than concrete conditions regardless of individuals’ (anti)egalitarian values. This suggests that egalitarian values help shape people’s attitudes but are less likely to impact their choices, even in more distal and abstract contexts. Furthermore, we did not find support for the idea that individuals’ choices to promote diversity more in the abstract versus in the concrete were driven by the salience of diversity pros versus cons. It could be that while *attitudes* towards diversity are sensitive to its salient pros, individuals’ *choices* to promote diversity are driven by other factors. In fact, this may be why, in both Studies 2 and 5, individuals’ positive attitudes towards diversity were not correlated with their choices to promote diversity and why (anti)egalitarian beliefs did not influence the impact of construal level on diversity choices in our final study. We also observed that when more variables were measured concomitantly (as in Studies 4 and 5), the effect of construal was stronger, or only present in one of them, often the diversity choices. While we found an impact of construal on the salience of diversity pros in Study 4 (where only diversity attitudes were measured), we did not find this effect in Study 5 (where diversity attitudes and choices were measured). More research is needed to disentangle what appears to be distinct cognitive mechanisms stemming from more abstract versus concrete construals that shape favorable attitudes towards choices to promote diversity.

### 7.3. Theoretical Implications

This research has important theoretical and practical implications for understanding why and when people have ambivalent views toward diversity. First, our research offers a potential explanation for the “double-edged sword” effect, namely that diversity operates in the workplace with both positive and negative outcomes (see [6]; [17]; [38] for reviews). These effects co-exist in organizations: diversity enhances decision-making and helps organizations to more effectively meet the needs of diverse clientele ([6]), but can also increase interpersonal conflict and reduce trust, communication, and coordination ([3]; [34]). We show that people can switch their cognitive focus on the positive and abstract aspects of diversity as they can stay anchored in the more concrete and less positive aspects of diversity, thus influencing their diversity attitudes and choices.

Second, our research contributes to recent debates about the impact and support of diversity practices ([10]; [24]). In an organizational and political context plagued with diversity backlash, it is important to offer conceptual and practical tools to organizations committed to implementing diversity initiatives. In line with research by [24] ([24]), our study also confirms that attitudinal and behavioral support are both important but rarely aligned. Our research suggests that people who lack attitudinal support can gain support by focusing more abstractly on the benefits of diversity, while people who lack behavioral support should focus more concretely on effective ways to manage diversity and thus overcome the feasibility concerns ([23]). Managers often resist diversity initiatives because their structural power position and organizational identification make them believe that their organization is equitable and that diversity initiatives are unnecessary ([42]). Our research suggests that managers must change from their abstract focus on the organization to a more concrete focus on workforce inequalities to support diversity initiatives. In addition, by considering individuals’ cognitive focus, tensions between advantaged and disadvantaged workers regarding DEI initiatives are susceptible to decrease ([22]; [30]). For example, a concrete cognitive focus on the benefits of inequality reduction might give the deserved recognition to the disadvantaged groups and might help the advantaged groups to reduce their ambivalent view.

Third, our research has implications for research on construal levels. Interestingly, while individuals generated more diversity pros in the abstract relative to the concrete construal conditions, which went on to shape positive diversity attitudes, the salience of diversity cons did not vary by construal level. This finding mirrors that of [29] ([29]), who found that while an abstract mindset increased attention on the pros of a risky action relative to a concrete mindset, there were no differences in the extent to which individuals focused on the cons of a risky action by construal level. Theoretically, this may suggest that because pros are superordinate to and more primary than cons, a focus on pros is more sensitive to shifts in construal level. Diversity complaints may not drive less favorable attitudes toward workplace diversity. Indeed, in our studies, individuals with egalitarian and anti-egalitarian beliefs did not differ in the number of diversity cons that they listed. This is in line with the study of [42] ([42]), which found that the biggest barrier within middle management to increasing organizational diversity might not be perceived costs but rather a general attitude of indifference in the face of inequalities. Our results about the prominence of diversity pros also partially challenge the results of [23] ([23]), who found that people might have feasibility concerns about supporting diversity in concrete situations. Our study, along with [42] ([42]), suggests that the desirability of diversity might still be a concern for many, especially for those with low egalitarian beliefs.

Fourth, our research suggests a limited impact of anti-egalitarian beliefs in the domain of diversity support. We show that people with egalitarian beliefs are likely to support diversity more as an abstract idea rather than as a real source of value within interactions in their own team, probably because of the congruence between their broad ideals of diversity underlying egalitarian beliefs and the abstract focus on the importance of diversity. In fact, when considering the impact of diversity concretely within their work contexts, these egalitarian individuals’ attitudes toward diversity look like those of their anti-egalitarian counterparts. This is consistent with the findings of [56] ([56]), who showed that multiculturalism had a positive impact in reducing White Americans’ prejudice toward ethnic minorities only when multiculturalism was framed abstractly, not concretely. Our results also align with studies on the “principle–implementation gap”, showing that people support abstract principles of diversity while simultaneously opposing concrete policies that help achieve such a goal ([9]).

### 7.4. Practical Implications

This research also has several practical implications. First, it suggests that managers should make people’s ambivalent views on diversity explicit and acknowledge its positive and negative effects in an organizational context. For example, in hiring committees, the managers should start by letting members express their opinions regarding a diverse hire and then acknowledge that both those who support and those who oppose the initiative might have good reasons for it. Then, the manager should encourage the committee members to think concretely about ways to welcome the new employees in a way that benefits the person and the organization. Second, managers should be able to activate contexts in which the abstract and concrete perspectives on diversity are made salient. For example, if the objective is to obtain support for a diversity initiative or to obtain workers’ engagement for working in diverse teams, then the focus should be on the more abstract and distal horizons with regard to diversity. However, if the objective is to implement diversity initiatives or form diverse teams in the short run, then the focus should be on the concrete benefits and the specific way in which diversity should be managed ([55]). While it is true that people in concrete contexts focus on the feasibility of diversity ([23]), our studies suggest that focusing on feasibility does not necessarily mean focusing on the cons of diversity, as it is often assumed. Third, managers should consider showing concrete data and facts about the benefits of diversity rather than displaying broad statements about diversity or counting on people’s egalitarian beliefs. People with egalitarian beliefs have stronger attitudinal support with diversity than those with anti-egalitarian beliefs, but only in abstract contexts. In concrete situations, their attitudes and behavior do not appear to differ from those with anti-egalitarian beliefs. Fourth, our findings suggest a potential lever for what practitioners have termed “the frozen middle”, whereby diversity initiatives do not meet resistance from leaders at the top of the organizational hierarchy but instead from middle managers ([37]; [42]). Individuals at the highest levels of leadership likely have a more socially distant view of how teams operate on the ground, prompting a more abstract (and more positive) view of diversity’s value (see also [32]). Middle managers closer to team operations may instead have a more concrete construal of diversity’s value that is relatively more negative. Therefore, one solution for middle managers is to set the default of diversity initiatives at the abstract level to obtain their support before implementing those initiatives concretely (see research on the choice architecture approach—[15]). Likewise, our findings shed light on cognitive processes occurring at the individual level that may contribute to instances of organizational decoupling more broadly, whereby there are gaps between idealistic organizational intentions and actual policy implementation ([7]). Managers can avoid this gap by setting diversity conversations, actions, and metrics at the same construal level, ideally at the concrete level, where employees can clearly understand what the organization wants to achieve with diversity, why, and how exactly.

## 8. Conclusions

Taken together, this work highlights why advancing diversity within organizations remains such a pervasive challenge. By utilizing construal level theory to examine diversity attitudes and choices, we contribute to growing bodies of knowledge in two items of literature, one that examines antecedents of diversity attitudes and support in the workplace and another that utilizes construal level theory to explore important outcomes of interest within organizational psychology and organizational behavior that may be used practically by managers. Furthermore, we contribute to management research on diversity, highlighting a cognitive mechanism for why those who generally support diversity (those with egalitarian beliefs) may fail to do so within their own organizational contexts. In contrast, previous research emphasizes the importance of targeting diversity initiatives to individuals most receptive to diversity’s aims. Our findings suggest that leaders may do well to also emphasize the benefits of diversity *concretely* within their particular company and employees’ own teams, rather than for business in general, as even egalitarian individuals’ attention to the positives of diversity is likely to fade as they think about the realities of diversity within their organizational contexts.

## Figures and Tables

**Figure 1 behavsci-15-00585-f001:**
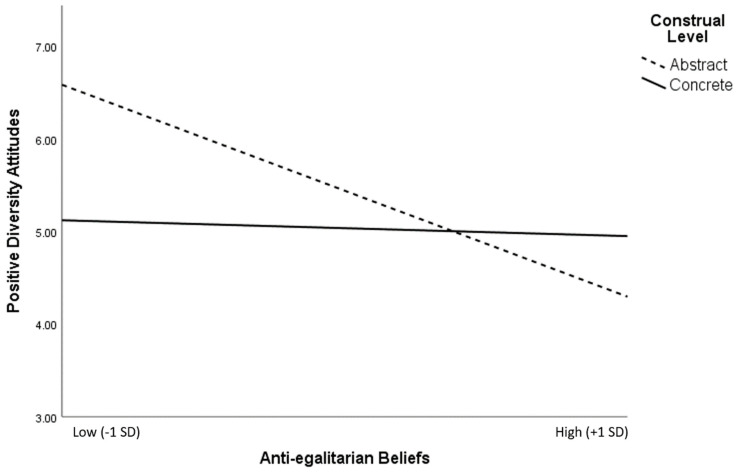
Study 3’s interaction between construal level and anti-egalitarian beliefs on positive diversity attitudes (1–7), plotted at +/−1 SD around the means on the continuous predictor.

**Figure 2 behavsci-15-00585-f002:**
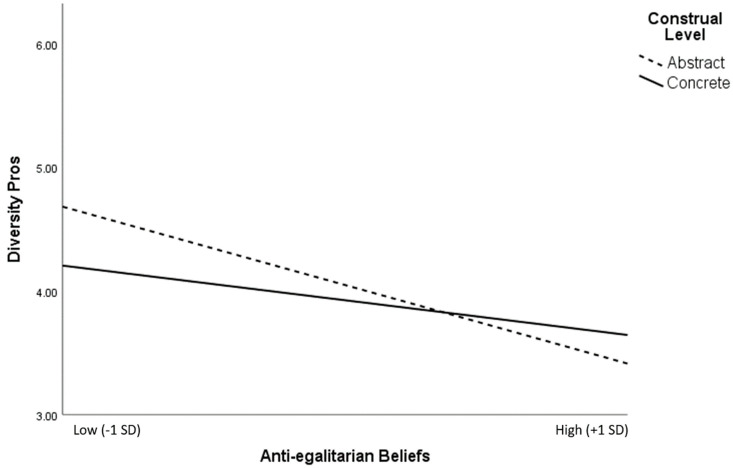
Study 4’s interaction between the construal level and anti-egalitarian beliefs on the number of diversity pros (0–10), plotted at +/−1 SD around the means on the continuous predictor.

**Figure 3 behavsci-15-00585-f003:**
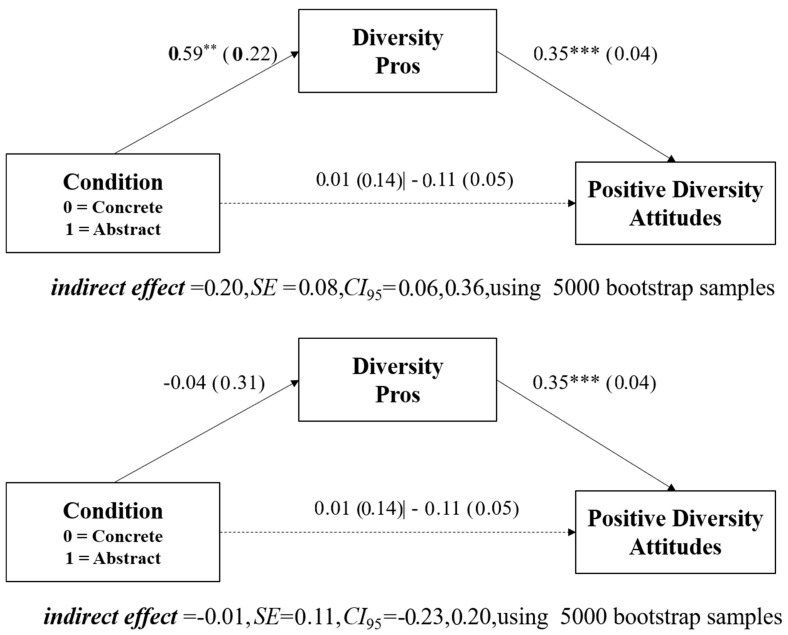
Top panel: Study 4’s indirect effect of construal level on positive diversity attitudes through diversity pros at the 16th percentile of anti-egalitarian beliefs. Middle panel: Study 4’s indirect effect of construal level on positive diversity attitudes through diversity pros at the 50th percentile of anti-egalitarian beliefs. Bottom panel: Study 4’s indirect effect of construal level on positive diversity attitudes through diversity pros at the 84th percentile of anti-egalitarian beliefs. Standard errors are shown in parentheses. ** *p* < 0.01. *** *p* < 0.001.

**Table 1 behavsci-15-00585-t001:** Overview of the method and hypotheses (for all studies).

	Sample	IV	DV	MO	Hypotheses Tested	Evidence for Hypotheses
**Study 1 (*n* = 61)**	Dutch students	Most vs. your groups	Diversity attitudes		H1a	Yes
**Study 2 (*n* = 70)**	Dutch workers	Most vs. your company and teams	Diversity attitudes Diversity choices		H1a	Yes
					H2a	Yes
**Study 3 (*n* = 95)**	White US workers	Most vs. your groups and teams	Diversity attitudes	SDO	H1a	Yes
					H1b	Yes
**Study 4 (*n* = 194)**	White US workers	Most vs. your groups and teams	Diversity attitudes Diversity pros and cons	SDO	H1a	No
					H1b	No
					H3a	Yes
					H3b	Yes
					H4a	Yes
					H4b	Yes
**Study 5 (*n* = 168)**	White US workers	Most vs. your groups and teams	Diversity attitudes Diversity choices Diversity pros and cons	SDO	H1a	No
					H1b	No
					H2a	Yes
					H2b	Yes
					H3a	No
					H3b	No

**Table 2 behavsci-15-00585-t002:** Means, standard deviations, and Pearson’s correlations among variables (for all studies).

	Mean	*SD*	1	2	3	4	5	6
Study 1 (*n* = 61)								
Positive Diversity Attitudes	4.80	0.90	(0.79)					
Study 2 (*n* = 70)								
Positive Diversity Attitudes	5.47	0.83	(0.75)					
2. Overall Diversity Choices	−13.19	3.40	0.09					
3. Demographic Diversity Choices	−9.84	2.74	0.01	0.78 **				
4. Cognitive Diversity Choices	−3.34	2.15	0.14	0.60 **	−0.05			
Study 3 (*n* = 95)								
Positive Diversity Attitudes	5.38	1.12	(0.85)					
2. Anti-egalitarian Beliefs	2.43	1.39	−0.64 **	(0.92)				
Study 4 (*n* = 194)								
Positive Diversity Attitudes	5.45	1.10	(0.81)					
2. Anti-egalitarian Beliefs	2.40	1.46	−0.57 **	(0.94)				
3. Diversity Pros	4.11	1.97	0.38 **	−0.29 **				
4. Diversity Cons	3.52	2.06	−0.07	0.07	0.58 **			
Study 5 (*n* = 168)								
Positive Diversity Attitudes	5.66	1.01	(0.86)					
2. Anti-egalitarian Beliefs	2.15	1.31	−0.65 **	(0.94)				
3. Diversity Pros	4.35	2.10	0.26 **	−0.26 **				
4. Diversity Cons	3.57	2.10	−0.03	0.00	0.60 **			
5. Overall Diversity Choices	6.87	4.46	0.13 ^ǂ^	−0.18 *	−0.00	−0.10		
6. Demographic Diversity Choices	6.17	3.03	0.13	−0.12	0.04	−0.09	0.69 **	
7. Cognitive Diversity Choices	0.70	3.21	0.06	−0.13	−0.04	−0.05	0.73 **	0.02

Scale reliability coefficients are shown in parentheses. ^ǂ^ *p* < 0.10. * *p* < 0.05, ** *p* < 0.01.

## Data Availability

The data presented in this study are available on request from the corresponding author due to privacy required in the project.

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
