# Peer review of "Diversity? Great for Most Just Less So for Me: How Cognitive Abstraction Affects Diversity Attitudes and Choices"

_behavsci, 2025, doi:10.3390/bs15050585_

Round 1
Reviewer 1 Report
Comments and Suggestions for Authors
1. What is the main question addressed by the research?
The main question addressed by this research, which is both important and timely, examines how levels of cognitive abstraction influence individuals' attitudes and decisions in favor of workplace diversity. The authors argue that individuals will hold more favorable attitudes toward the value of diversity and will more frequently make choices that enhance workplace diversity when they consider it more abstractly, that is, at a global and decontextualized level. They also emphasize the importance of understanding how individuals evaluate the positive and negative factors associated with diversity, as attitudes and decisions about diversity are central to enabling organizations to realize the benefits of their diversity initiatives fully.
2. What parts do you consider original or relevant to the field? What specific gap in the field does the paper address?
The article's introduction highlights the importance of individual attitudes toward diversity in achieving organizational goals in this domain. The authors adopt an interesting approach by using construal level theory to explain how individuals evaluate the advantages and disadvantages of diversity, while also incorporating the role of egalitarian beliefs as a moderator. This offers an original and relevant perspective to the field of organizational diversity, which has historically focused on structural or social approaches. I would like to emphasize that this is, in my view, the study's main potential contribution. While the literature on diversity is extensive, the approach adopted by the authors, which combines these elements to analyze attitudes and choices related to diversity, makes a highly relevant contribution to the field of organizational diversity.
However, the article does not position itself explicitly in relation to recent literature. The authors do not clearly identify the gaps that their work aims to address. Nevertheless, it is evident that their approach addresses unresolved underlying questions, particularly the limited application of cognitive frameworks to explain attitudes toward workplace diversity. Similarly, the role of egalitarian beliefs as a moderating factor in perceptions of diversity and the emphasis on cognitive mechanisms, such as the salience of the advantages of diversity, could provide valuable contributions but need to be better anchored in current discussions in the literature and explicitly mentioned.
These readings could maybe be helpful:
Waldrop, R. J., & Warren, M. A. (2024). Exploring Egalitarianism: A Conceptual and Methodological Review of Egalitarianism and Impacts on Positive Intergroup Relations. Behavioral Sciences, 14(9), 842. https://doi.org/10.3390/bs14090842
Nguyen, Tina, and Kentaro Fujita, 'On Psychological Distance and Construal Level: A Regulatory Perspective', in Donal E. Carlston, Kurt Hugenberg, and Kerri L. Johnson (eds), The Oxford Handbook of Social Cognition, Second Edition, 2nd edn, Oxford Library of Psychology (2024; online edn, Oxford Academic, 21 Aug. 2024), https://doi.org/10.1093/oxfordhb/9780197763414.013.13
3. What does it add to the subject area compared with other published material?
The article examines attitudes and choices related to diversity by using construal level theory, which lends it a degree of originality. However, when compared with other published works, its specific contribution remains difficult to fully assess due to a lack of contextualization within the recent literature. The authors mention intriguing cognitive mechanisms, such as the salience of diversity's advantages, but they do not sufficiently discuss how these findings distinguish themselves from or align with existing similar studies.
Moreover, while the cognitive approach is relevant, it does not clearly address emerging questions in the field, such as the evolution of diversity perceptions in rapidly changing organizational contexts or the impact of new societal dynamics. The study could have gained depth by incorporating explicit comparisons with alternative frameworks or theories, such as models of organizational justice or inclusive leadership, to situate its contribution better.
Finally, the lack of concrete applications for the results poses a challenge. Although the theoretical approach is innovative, the absence of clear practical recommendations for managers or decision-makers limits the article’s potential impact in the field of diversity management. A more developed discussion on how the findings could inform organizational policies would have better articulated its relevance to the subject area.
4. Are the conclusions consistent with the evidence and arguments presented? Were all the main questions posed addressed? By which specific experiments?
The article relies on five studies to test its hypotheses, which is one of its major strengths, given the richness and diversity of the data produced. The results of the studies provide interesting insights and generally seem to support the arguments put forward. However, the exploitation of the data remains both insufficient and fragmented, which limits the coherence of the conclusions.
The article presents four main hypotheses and several sub-hypotheses, but their treatment across the studies lacks clarity and integration. Each study is described in isolation, and the final discussion is too brief to offer a cohesive overview. The conclusions, while interesting, do not explicitly revisit the initial hypotheses, leaving the reader uncertain about how each study contributes to validating or invalidating the various hypotheses. This disconnect between the hypotheses, studies, and conclusions undermines the readability and analytical impact of the article.
To enhance the article’s coherence and impact, it would be beneficial to introduce a global and integrated perspective on the studies and their contributions. For instance, an initial table could cross-reference the hypotheses, the studies, and their protocols. In the discussion, this table could be revisited to indicate the main results obtained and the answers provided to the hypotheses. Such a presentation would give readers an overarching view and make it easier to understand how the different parts of the article fit together.
As an indicative example, I suggest two articles from the journal Current Psychology. These papers present fewer studies but, in my opinion, succeed in better integrating them to provide a cohesive and articulated whole:
Cuadrado, I., Estevan-Reina, L., López-Rodríguez, L. et al. To be or not to be egalitarian, that is the question: Understanding the complexity of ethnic prejudice in the workplace. Curr Psychol, 42, 18033–18051 (2023). https://doi.org/10.1007/s12144-022-02872-y
Hung, YS., Lo, SY. Competence or morality? Investigating how psychological distance moderates individuals’ attitudes toward organizations’ behavioral ambivalence. Curr Psychol, 43, 16499–16513 (2024). https://doi.org/10.1007/s12144-023-05560-7
I sincerely believe that the authors possess valuable data and rely on a relevant framework. By moving away from a narrative approach and adopting a more analytical stance, the article could make a significant contribution.
5. Are the references appropriate?
The references cited in the conceptual framework are generally relevant and come from established works in the fields of social psychology and organizational diversity. The authors draw on key sources, such as foundational studies on construal level theory and workplace diversity research. However, some gaps appear in the integration of more recent references, particularly in related fields. While the choice of construal level theory seems relevant, readers may need a clearer explanation of why this specific framework was chosen over other, more traditional conceptual frameworks.
Moreover, although the article relies on relevant theoretical frameworks, it could have been enriched with more recent studies to situate its contributions within the current literature better. For instance, recent studies exploring the impact of societal dynamics on attitudes toward diversity or comparable empirical analyses could have strengthened the article's theoretical and empirical foundations. This is especially pertinent given the rapidly evolving perceptions and practices surrounding diversity in organizations.
Finally, the discussion appears to be the manuscript's major weakness. As mentioned earlier, the discussion does not adequately highlight the key learnings in a solid and integrated manner, nor does it propose actionable insights. Furthermore, the discussion largely overlooks the existing literature in two key areas: situating the contributions within the literature and demonstrating whether the literature supports these conclusions or if the findings represent genuinely new knowledge. A thorough return to the literature seems necessary for revising this work.
Here are a few studies that might enrich the bibliography of this work:
Brasseur, M. (2012). Les croyances sur la diversité et leurs rôles dans le management. La Revue des Sciences de Gestion, N° 257(5), 71-79. https://doi.org/10.3917/rsg.257.0071
Garg, S., & Sangwan, S. (2021). Literature review on diversity and inclusion at workplace, 2010–2017. Vision, 25(1), 12-22.
Roberson, Q. M. (2019). Diversity in the workplace: A review, synthesis, and future research agenda. Annual Review of Organizational Psychology and Organizational Behavior, 6(1), 69-88.
Seliverstova, Y. (2021). Workforce diversity management: A systematic literature review. Strategic Management-International Journal of Strategic Management and Decision Support Systems in Strategic Management, 26(2).
Yadav, S., & Lenka, U. (2020). Diversity management: a systematic review. Equality, Diversity and Inclusion: An International Journal, 39(8), 901-929.
I sincerely hope the authors will find the time and determination to revise this manuscript, which I believe has significant promise.
Reviewer 2 Report
Comments and Suggestions for Authors
The authors ask questions of theoretical and practical importance. Moreover, I found the paper, especially the introduction, to be well-written. Unfortunately, I found that issues with the measures and manipulations (detailed below) muddied any strong conclusions I could take away from the presented studies. Further, I found the paper would have really benefitted from pre-registering the included studies to bolster confidence in hypotheses, excluded measures, participants, and conditions, and analytic choices (e.g., Diversity Choices, pro’s and con’s models). Below I detail specific points of feedback, with the more significant points raised first.
One methodological challenge in Study 1 & 3 is ensuring people are thinking of comparable groups inasmuch as, in both conditions, they are thinking about groups that would just as much be the types of groups that would benefit from diversity. In the world of possible groups they could be thinking of in response to the prompts, it might be the case that larger groups or perhaps groups that serve a function (rather than, e.g., a peer group, a family group, an ethnic group) are those for which people might think that diversity provides value. However, for groups they are likely to identify with, to think of as the groups they belong to, these groups could be of the sort that diversity is not thought to be as beneficial (e.g., a family group) or is not practical/possible (e.g., an ethnic group). Particularly if people are imagining their groups as ethnic groups, then the items take on a different meaning in ways that detract from the purpose of this inquiry. For instance, “I believe that groups to which I belong benefit from the involvement of people with different backgrounds” could be read paternalistically (i.e., other groups help or provide assistance to my group), or “I believe it can be very problematic if groups to which I belong include different people” could be read as a referendum of who counts as Dutch or, in the case of White Americans prompt concerns a la “replacement theory”. It might be that the adjustment in the prompt to read “groups and teams” in Study 4 addressed this. By naming “teams” in the measure, participants in the concrete condition may have been nudged to think of groups that were more comparable to those thought of by participants in the abstract condition. This might be an explanation for why the hypothesized effects were not observed in Study 4.
Studies 2 & 5’s measures address the above concern, though the instructions to participants seems to create a confound. In the abstract condition, they are instructed to form a team according to how the company should do it. In the concrete condition, instead of being asked to form their own team according to how it should be formed, they are tasked with forming a team that they would feel most comfortable working with. As such, it’s not just the construal level that’s varying, but the principle by which participants are told to construct the team. As such, we cannot discern what change is driving the condition effect.
Further, with this measure, it would seem that not every participant would have the same potential range of scores given their own characteristics and the distribution of characteristics among the potential team members. If it is the case that every individual has the same theoretical range of values, despite their personal demographics, it would be helpful to have that explained. If not, it seems the authors would want to adjust for that in their analyses. I also had a couple of reservations about the profiles themselves. For one, including level of education could make interpretation here tricky since it is a bona fide occupational qualification. If participants see higher levels of education as the factor companies would select on (as a bona fide occupational qualification), then they might be pushed into selecting those individuals with higher levels of education given their expected performance as highly qualified employees. All of the profiles with at least a master’s degree are non-White. I’m assuming this is different from the majority of participants in both Studies 2 & 5 (also, it would be helpful to have participant demographics reported here, especially in Study 2), and it could be (at least in part) driving these effects. Also, the last profile of someone of “African America descent” who grew up in the Netherlands is quite unusual in ways that make me want to know more. All of the other participants’ descent was consistent with where they grew up. This person’s was not, and happened to be listed as growing up in the same place as Study 2’s participants. This, paired with such an unusual description of someone being of African American descent but growing up in the Netherlands (England in Study 5 may have had an unusual impact on participants, selecting her more or less often, that it would be good to know whether results are robust to the inclusion of this profile. Also, which trait is matched to participants? The ethnic background or nation of residence?
In Study 4, it’s not clear why the regression models assessing the interaction between condition and anti-egalitarian beliefs on salience of pro’s and con’s controls for the one when assessing the impact on the other. It would be helpful to see if the findings hold without that control.
In Study 3 (and others), a system justification measure is referred to as a measure included for exploratory purposes. It seems like the expectations regarding system justification would be quite similar to those regarding SDO. As such, it would be important to report results with system justification as well and to interpret any differences than what was observed with SDO.
Round 2
Reviewer 1 Report
Comments and Suggestions for Authors
The revised manuscript presents several significant improvements in response to the previous review. The authors have strengthened the theoretical positioning by incorporating recent and relevant references, particularly regarding the role of cognitive abstraction in diversity attitudes. The integration of new studies provides a more comprehensive contextualization within the broader literature on organizational diversity. The discussion now offers a more structured and cohesive synthesis of the findings, and the addition of a section on practical implications is a valuable contribution.
Despite these improvements, some issues remain that warrant further clarification. One of the key concerns is the inconsistency in results across studies. While some experiments confirm the effect of construal level on diversity attitudes, others fail to replicate this pattern. Studies 4 and 5, in particular, do not support the direct influence of cognitive abstraction on diversity attitudes, which weakens the robustness of the overall conclusions. Although the authors acknowledge these discrepancies, their explanations remain somewhat superficial. It would be beneficial to provide a more detailed and theoretically grounded justification for these inconsistencies, perhaps by exploring alternative mechanisms or considering potential confounding variables that may have influenced the results.
Another issue concerns the operationalization of the proposed mediating mechanism. The authors suggest that egalitarian beliefs play an essential role in shaping diversity attitudes depending on construal level. However, the empirical findings indicate that this effect is absent in concrete decision-making contexts. This raises concerns about whether egalitarian beliefs function as a meaningful moderator in this framework. If individuals with egalitarian beliefs do not exhibit stronger support for diversity in concrete situations, it challenges one of the study’s fundamental assumptions. The authors should clarify whether this suggests a boundary condition to their theory or whether alternative explanations might account for this unexpected result. If I have misunderstood the authors' reasoning on this point, I would appreciate further clarification to better grasp how this mechanism operates in their framework.
The practical implications section represents an important addition, but it remains somewhat underdeveloped. The recommendations for practitioners are framed in broad terms and lack specificity. For instance, stating that managers should address cognitive ambivalence towards diversity does not provide concrete guidance on how this can be effectively achieved in organizational settings. A more detailed discussion of actionable strategies, perhaps drawing on evidence from diversity training programs or cognitive interventions, would strengthen the practical relevance of the study.
Fasolo, B., Heard, C., & Scopelliti, I. (2024). Mitigating Cognitive Bias to Improve Organizational Decisions: An Integrative Review, Framework, and Research Agenda. Journal of Management, 0(0). https://doi.org/10.1177/01492063241287188
Cansın Arslan et al. ,Behaviorally designed training leads to more diverse hiring.Science387,364-366(2025).DOI:10.1126/science.ads5258
Given these remaining concerns, a minor revision is warranted to refine the theoretical justifications for the inconsistencies in results, clarify the role of egalitarian beliefs, and enhance the practical implications. Taking these points into account would considerably strengthen the contribution of this research and ensure that the results are interpreted with the necessary rigor. The authors are also encouraged to provide a more robust justification for the observed variations in results and to refine their discussion to better align theoretical claims with empirical findings. I guess the manuscript is now in a strong position, and with these final refinements, it appears very close to reaching its full potential.
Comments on the Quality of English LanguageAlthough the overall readability of the manuscript has improved, there are still areas where clarity could be enhanced. Some paragraphs, particularly in the discussion section, contain unnecessary repetitions or complex phrasing that could be streamlined for better comprehension. Additionally, minor typographical errors persist throughout the text, including spelling mistakes such as "bystandaers" instead of "bystanders" and "fesability" instead of "feasibility." A careful proofreading would help eliminate these errors and further improve the manuscript’s clarity. (I suggest that the final proofreading in English be entrusted to a third party to ensure linguistic accuracy and clarity.)
Author Response
Thank you for the positive and constructive feedback and for the possibility of revising the manuscript.
Comment 1: One of the key concerns is the inconsistency in results across studies. While some experiments confirm the effect of construal level on diversity attitudes, others fail to replicate this pattern. Studies 4 and 5, in particular, do not support the direct influence of cognitive abstraction on diversity attitudes, which weakens the robustness of the overall conclusions. Although the authors acknowledge these discrepancies, their explanations remain somewhat superficial. It would be beneficial to provide a more detailed and theoretically grounded justification for these inconsistencies, perhaps by exploring alternative mechanisms or considering potential confounding variables that may have influenced the results.
Response 1: Thank you for pointing this. We agree that the lack of support for the effect of construal on diversity attitudes in the last two studies weakened a bit the overall conclusion. However, we do not believe that the problem is theoretical; rather, it is methodological. In those last two studies DVs are measured with a counterbalanced order. As it is now stated at page 31, this could have complexified the design of those studies, where the influence of one variable is mixed with another. For example, in Study 4, where diversity attitudes and diversity pros were measured concomitantly, the effect of construal of diversity pros might have diluted the effect on diversity attitudes. In other words, when we assessed the mediator, the effect of construal on the diversity attitudes disappeared, suggesting that the indirect impact canceled out the presence of the direct effect. A similar explanation is plausible for Study 5 where, this type, the measure of diversity choices seems to have canceled the effect on the other DVs.
We do believe that, theoretically, our hypotheses are sound and in line with existing literature. Therefore, we only provided methodological explanations.
Comment 2: Another issue concerns the operationalization of the proposed mediating mechanism. The authors suggest that egalitarian beliefs play an essential role in shaping diversity attitudes depending on construal level. However, the empirical findings indicate that this effect is absent in concrete decision-making contexts. This raises concerns about whether egalitarian beliefs function as a meaningful moderator in this framework. If individuals with egalitarian beliefs do not exhibit stronger support for diversity in concrete situations, it challenges one of the study’s fundamental assumptions. The authors should clarify whether this suggests a boundary condition to their theory or whether alternative explanations might account for this unexpected result. If I have misunderstood the authors' reasoning on this point, I would appreciate further clarification to better grasp how this mechanism operates in their framework.
Response 2: Thank you for this, but here, there is a bit of misunderstanding. Egalitarian beliefs are not our proposed mechanism; it is a moderator factor that seems to reinforce (Study 4) or only makes appear the effect of construal in the abstract condition (Study 3). This is not at all surprising. As mentioned on page 10, only people who give some value to diversity can be impacted by the construal level. The ones who are opposed will continue to be opposed regardless of the construal. At the same time, the fact that egalitarian beliefs have a positive impact only in the abstract condition is in line with the existing literature. To clarify this, we state on page 11: "This also suggests that the role of anti-egalitarian beliefs should be stronger in the abstract than in the concrete construal, given that the abstract focus leads to more inclusive categorization, support for multiculturalism, and more proc-social behaviors, which correspond to what people with egalitarian beliefs generally support (McCrea et al., 2012; Yogeeswaran & Dasgupta, 2014). "
We also clarify this aspect in the discussion at page 35: "This is consistent with Yogeeswaran and Dasgupta (2014)’s findings who showed that multiculturalism had a positive impact in reducing White Americans’ prejudice toward ethnic minorities only when multiculturalism was framed abstractly, not concretely. Our results also align with studies on the “principle-implementation gap,” showing that people support abstract principles of diversity while simultaneously opposing concrete policies that help achieve such a goal (Dixon et al., 2010)."
Comment 3: The practical implications section represents an important addition, but it remains somewhat underdeveloped. The recommendations for practitioners are framed in broad terms and lack specificity. For instance, stating that managers should address cognitive ambivalence towards diversity does not provide concrete guidance on how this can be effectively achieved in organizational settings. A more detailed discussion of actionable strategies, perhaps drawing on evidence from diversity training programs or cognitive interventions, would strengthen the practical relevance of the study.
Response 3: Thank you for this useful suggestion. Based on your references, we have developed the practical implications to include more actionable recommendations. For the first proposed implications (recognizing the ambivalence of diversity), we now give an example of how this can be discussed and implemented in hiring situations.
Page 35: "For example, in hiring committees, the managers should start by letting members express their opinions regarding a diverse hire and then acknowledge that both those who support and those who oppose the initiative might have good reasons for it. Then, the manager should encourage the committee members to think concretely about ways to welcome the new employees in a way that benefits the person and the organization. "
In addition, based on Fasolo et al (2024)'s paper about the choice architecture, we propose to set the default diversity initiatives at the abstract level for the middle managers to get their support before asking them to implement them concretely (also on page 36).
To reduce the implementation gap, we also propose to set diversity talk, actions, and metrics in a coherent way, ideally at the concrete level. Page 36: "Managers can avoid this gap by setting diversity talk, actions, and metrics at the same construal level, ideally at the concrete level where employees can clearly understand what the organization wants to achieve with diversity, why, and how exactly."
Thank you again for your useful comments. We also addressed the typos we identified.
We hope you’ll find this version improved and close to the publication phase.